# Seafood Sustainability Supply Chain Trends and Challenges in Japan: Marine Stewardship Council Fisheries and Chain of Custody Certificates

**Minako Iue [1,2,*], Mitsutaku Makino [3] and Misuzu Asari [1]**

[1] Graduate School of Global Environmental Studies, Kyoto University, Kyoto 606-8501, Japan
[2] Sailors for the Sea Japan, Yokohama 236-8648, Japan
[3] Atmosphere and Ocean Research Institute, Tokyo University, Tokyo 192-0397, Japan
[*] Correspondence: minako-i@sailorsforthesea.jp

**Abstract:** In Japan, fisheries' product has decreased since the 1980s, and the government amended the fisheries law in 2020 to shift to managed fisheries. However, awareness of seafood sustainability is still low. This study identifies the reason for the low awareness and states the necessary actions to increase sustainable seafood consumption. The proactive stakeholders in seafood supply were surveyed to determine the current status of sustainable seafood sales. Marine Stewardship Council-certified fishers and MSC's Chain-of-Custody certification holders answered the questionnaires. Fishers showed a positive attitude, citing proactive sales channel expansions and contributions to sustainability. Distributors were primarily passive, mainly because they obtained the certification at the request of their clients, and about half of them did not disseminate information between consumers and suppliers. Both fishers and distributors expect the government to promote campaigns and education. The stages of the awareness of producers, intermediary trade, and retailers are different and did not align. Therefore, if the supply chain stakeholders proactively educate themselves, choose sustainable products from the fishers, and pass the sustainability information to the consumers, sustainable seafood consumption would progress.

**Keywords:** sustainable fishery; sustainable seafood trade; supply chain; fisheries certificate; CoC certificate

## 1. Introduction

With the spread of the Sustainable Development Goals (SDGs) concept, sustainability perspectives are being sought for food to address issues such as hunger and food loss, and to achieve carbon neutrality. Sustainable seafood management is one of them [1–3]. For marine environments, various policies, methods, and market mechanisms are being explored to ensure the appropriate use of marine resources by humans. In Japan, such efforts have just begun.

The level of more than half of the fishery resources in Japan is considered to be low, and, in December 2020, the Japanese government enacted the "reformed fisheries policies". This amendment is a fundamental revision of the Fishery Law, which was enacted in 1949 [4]. Additionally, the "Act on the Optimization of Domestic Distribution of Specified Aquatic Animals and Plants", the so-called anti-illegal, unreported, and unregulated (IUU) law, was passed in the same month. With these developments, Japan entered a major turning point toward achieving sustainable fisheries and seafood consumption. Following the EU and the USA, the catch documentation scheme will be mandated, which enables the sustainable and effective use of resources and eliminates IUU fisheries. Furthermore, Japan is expected to take international responsibility in contributing to the sustainable development of the fisheries industry in Asia in the future, based on the wisdom and experience of the Japanese-style co-management approach [5].

Japanese private sectors have also promoted sustainable seafood consumption to join the international trend. The Marine Stewardship Council (MSC), internationally certifying sustainable fishers, increased from 2 to 10 fisheries in Japan between 2015 and 2021. The number of distributors that obtained its Chain-of-Custody (CoC) certification has also increased from approximately 80 to 300 companies [6,7]. As of 2022, 14 domestic fisheries have been approved by the Aquacultural Stewardship Council (ASC) for farmed fish, and 166 companies have obtained CoC certification [8]. Marine Ecolabel (MEL), a Japanese fisheries certification, tightened its evaluation criteria to international standards and released Version 2, which was approved by the Global Sustainable Seafood Initiative (GSSI) in December 2019, as were MSC and ASC [9]. Sailors for the Sea Japan (SFSJ), the owner of Japan's sustainable seafood rating program Blue Seafood Guide (BSG), became a charter member of the Global Seafood Ratings Alliance. SFSJ has accelerated its activities by establishing comprehensive agreements with local governments, including the Tokyo Metropolitan Government and Mie Prefecture, to create their local version of BSG, and partnered with over 60 companies [10]. Furthermore, three major fishery companies, Nissui, Maruha Nichiro, and Kyokuyo, have joined the Seafood Business for Ocean Stewardship (SeaBOS), an international organization for dialogue and initiatives between scientists and businesses. The movement to promote sustainable seafood has begun to form.

Public awareness and market growth can be accelerated through international official events. In the UK, the procurement of sustainable seafood increased in the wake of the London Olympics [11,12]. The concept of sustainable seafood was seen at Japan-led international events, such as the Tokyo 2020 Olympic and Paralympic Games (Tokyo 2020). At the Eighth Pacific Island Summit in Fukushima in 2018, the Spouse Program symposium on ocean conservation was held [13]. The G20 Osaka Summit 2019 included ending IUU fisheries according to the Summit declaration, in addition to the announcement of Osaka Blue Ocean Vision and the Spouse Program, which followed the same theme [14]. The wife of the Prime Minister hosted the ocean conservation forum at the 2019 United Nation General Assembly [15]. However, these efforts have not yet had an impact on consumer awareness.

A survey showed that less than 10% of Japanese consumers are aware of the fishery ecolabel [16]. Hori et al. [17] showed that an environmentally conscious purchasing behavior model for seafood products is structured as follows: social norm evaluation becomes the base variable, influencing evaluations of performance, effectiveness, and feasibility, which become mediating variables, influencing willingness to purchase, and finally leading to purchasing practices. Enhancing social norm evaluation is important in promoting sustainable seafood purchasing behavior. Although the number of MSC-certified fishers is increasing, only approximately 10 out of the approximately 79,000 fishing companies in Japan are currently certified [18]. Ecolabeling is less prevalent in Asian seafood markets than those in Europe and North America [19].

The purpose of this study is to identify why awareness of sustainable seafood consumption is low in Japan and to propose necessary actions to increase the demand for sustainable seafood. Since the previous studies on consumer awareness regarding sustainability do not address this question adequately, the paper looks at the different stages of stakeholder awareness in the supply chain associated with the seafood market. Therefore, the research hypothesis is that the stages of stakeholder awareness in the supply chain (producers, intermediary trade and retailers) are different and are not aligned, so the supply chain does not function to supply sustainable seafood or pass on sustainability information and the value of sustainable products. Therefore, it is vital to enforcing sustainable trade through a transparent and traceable seafood supply chain, realized by the further application of sustainability regulations to the supply chain and the collaboration of stakeholders. If the government supports the sustainable seafood trade, and supply chain stakeholders proactively educate themselves, choose sustainable products from the fishers, and pass the sustainability information to the consumers, sustainable seafood consumption would progress.

This paper categorized supply chain stakeholders into three groups—1. producers (fishers), 2. upstream distributors (manufacturers and intermediaries), and 3. downstream distributors (sales and restaurants)—and surveyed their respective attitudes.

## 2. Literature Review

Is sustainable seafood consumption achievable in Japan? The results of a survey of consumer preferences in Japan by Uchida et al. [20] showed a strong preference for domestic and natural seafood. Wakabayashi [21] compared farmed and wild fish in his survey: 70% of the respondents felt that wild fish tasted better, and twice as many people thought wild fish were safer. The connotations surrounding farmed fish were negative, with concerns regarding residual chemicals, marine pollution, excess fat, and a different texture and bad odor. In addition, Japanese food culture places importance on seasonality [22]. In response to this Japanese preference for domestic products, natural fish, and seasonality, and the demand for information, the Japanese government now requires labels to indicate the country of origin, whether fish are wild or farmed, and whether the product is raw or frozen [23]. According to FAJ statistics [24], consumer requirements for purchasing seafood products include taste, price, freshness, safety, seasonality, nutrition, and health. In addition, according to a survey by the Japan Fisheries Association [25], the key factors that led consumers to eat fish were nutritional intake, such as DHA, EPA, and protein; enjoying the season; and the convenience of processed products. The reasons for avoiding seafood included health-hindering effects such as anisakis, mercury, radiation, and microplastics, and the difficulty of eating fish due to their troublesome bones. The results of a survey by Maruha Nichiro [26] showed the following reasons for avoiding seafood: hard to eat or prepare, generating waste, hard to clean up, and smelly. However, none of the above statistics mentioned seafood sustainability.

Regarding sustainable seafood purchase behavior, Uchida et al. [27] suggested that consumers are more willing to pay for sustainable seafood when informed of the global seafood crisis. Hori et al. [28] also concluded that Japanese consumer acceptance of the importance of seafood sustainability has increased from 2017 to 2019, although progress has been limited. The results of the MSC [29] survey also showed that, in Japan, the proportion of people who say that they should switch to more sustainable seafood has increased from 49% in 2020 to 61% in 2022. These studies show an increase in consumer demand. Uchida et al. [27] reported that consumers respond to ecolabels only after receiving information on environmental issues, and that consumers show a general lack of awareness. As stated by Togawa [30], consumer decision-making structures are highly complex and large; measures that focus on sustainability alone and expect a change in consumer attitudes are not considered sufficient. Swartz et al. [31] found that the obstacles to seafood certification in Japan are due to a structural mismatch between the certification system and the Japanese domestic fisheries and seafood supply chain.

Pro-sustainable seafood behavior has been studied in western countries. Researchers have found that well-educated, young, and female populations is most likely to be proactive in sustainable seafood consumption. Asche and Bronnmann [32] found that attitudes toward ecolabels vary by market and consumer group. Wessells et al. [33] found that preferences for ecolabels exist in the USA, but vary by race, region, consumer group, and even certification body. Brécard et al. [34] found that a sample of European consumers revealed a tendency for well-educated young women to be the largest market for ecolabeled seafood. Salladarré et al. [35] also found that young, well-educated consumers preferred ecolabeled seafood products in the French market.

Gutierrez and Thornton [36] asserted that seafood ecolabel markets are not driven by consumer demand but by the interactions of social movement, organizations, states, consumers, and companies. Barclay [37] called the sustainable seafood movement a governance concert, where consumers act in concert with other types of actors along the supply chain. Pauly [38] stated that by combining strong legislation mandating the rebuilding of depleted stocks, institutions capable of implementing such legislation, the non-use of

destructive fishing methods such as trawls, and the establishment of networks of marine reserves in all countries, it should be possible to set fisheries on a sustainable course. Hilborn explained that an ecological focus alone does not guarantee long-term sustainability of any form, and that a socio-ecological perspective is essential to seafood sustainability if it is to be effective across cultures and in the future [39]. Socio-ecology describes a sustainable society in which humans and nature coexist in harmony; humans' moderate use of nature is considered essential [40,41]. The summary of the clarifications by the previous studies and the research gaps are shown in Table 1.

**Table 1.** Authors contribution table.

| Clarifications by the Previous Studies | Research Gaps |
|---|---|
| Previous studies for the motivation of seafood consumption: Strong preference for domestic and natural seafood. (Uchida et al., 2014 [20]) Prefer natural fish than farmed (Wakabayashi, 2011 [21]) Taste, price, freshness, safety, seasonality, nutrition, and health. Additionally, nutritional intake and season (FAJ 2022, [22]. JF, 2022 [25]) Raising awareness for seafood sustainability: Consumers are more willing to pay for sustainable seafood when informed of the global seafood crisis (Uchida et al., 2013 [27]) Acceptance of the importance of seafood sustainability has increased from 2017 to 2019, (Hori et al., 2020 [28]) Consumers respond to ecolabels only after receiving information on environmental issues, and that consumers show a general lack of awareness. (Uchida et al., 2013 [27]) The obstacles to seafood certification in Japan are due to a structural mismatch between the certification system and the Japanese domestic fisheries and seafood supply chain. (Swartz et al. 2019 [31]) A sample of European consumers revealed a tendency for well-educated young women to be the largest market for eco-labeled seafood. (Brécard et al. 2009 [34]) Seafood ecolabel markets are not driven by consumer demand but by the interactions of social movement, organizations, states, consumers, and companies. (Gutierrez and Thornton, 2014 [36]) An ecological focus alone does not guarantee long-term sustainability of any form, and that a socio-ecological perspective is essential to seafood sustainability if it is to be effective across cultures and in the future. (Hilborn et al., 2015 [39]) | Motivation and hurdles for obtaining MSC and CoC certificate in Japan. Perspective of supply chain. Mind of CoC holders. Differences in the level of awareness of proactive producers and distributors. Whether there is consistent, mutual distribution of sustainability information among the certified stakeholders. Whether the MSC and CoC certification is fully used, despite the additional effort and cost. Where the certificate holders find hurdles: internally and externally. |

## 3. Fish Stock Decline and the Definition of Sustainable Seafood

Marine fish stocks have declined globally over the last half century, in part due to overfishing and IUU fisheries. The percentage of fish stocks that are within biologically sustainable levels has declined from 90% in 1974 to 65.8% in 2017 [42]. The status of fish stocks in the Japanese Exclusive Economic Zone (EEZ) is worse than the global average. The Japanese Fisheries Agency (FAJ) reported in 2020 that only 23% of evaluated stocks were at a high level, and 53% were at a low level [10].

Sustainable Seafood has been defined by the multiple organizations, based on the Code of Conduct of Responsible Fisheries issued by the Food and Agriculture Organization of the United Nations in 1995.

The first and the most notable principles of sustainable fisheries were developed by the World Wildlife Fund and Unilever for MSC certificate, indicating three pillars: 1. stock status of target species, 2. ecosystem conditions, and 3. governance/management conditions [43]. Both fisheries' certificates and the consumer guides for sustainable seafood consumption provide definitions of sustainability. For example, Seafood Watch, a famous seafood rating program, evaluates seafood sustainability based on five criteria: 1. inherent

venerability, 2. status of stocks, 3. nature of bycatch, 4. habitat and ecosystem effects, and 5. management effectiveness. Iue et al. provided a comparative study of the criteria of the three programs—MSC, Seafood Watch and Blue Seafood Guide—a Japanese sustainable seafood rating program and showed that the style of categorizations are different but the elements of those three programs are shared. All three programs cover the following twelve evaluation criteria: 1. target stock management, 2. non-target stock management, 3. endangered, threatened and protected species management, 4. habitat management, 5. ecosystem management, 6. target stock status, 7. non-target stock status, 8. endangered, threatened and protected species status, 9. habitat status, 10. ecosystem status, 11. fishery management system, and 12. governance [10].

## 4. Materials and Methods

### 4.1. Questionnaire Survey for Producers

The questionnaires were administered to seven MSC-certified fishery companies, which covered all the MSC fisheries certified in Japan as of 2021. All of them answered were answered using Google Forms from 14 to 24 January 2022. The survey asked nine questions, including the reasons for taking on the challenge of MSC certification, the difficulties encountered, the reactions of consumers and business partners after certification, the impact of the Tokyo 2020 and SDGs, gains and losses from certification, and future challenges for MSC certification (see questionnaire sheet in Appendix A).

### 4.2. Questionnaire Survey for Distributors

A questionnaire survey was conducted on businesses (322 companies, answered by 98) throughout Japan that have obtained MSC/ASC CoC certification from 17 February to 19 March 2022, in the same manner as the questionnaire for fishers. The questions covered 22 items, including the purpose of obtaining CoC certification, the difficulties encountered, the reactions of consumers and business partners after obtaining certification, the impact of Tokyo 2020 and SDGs, gains and losses from certification, PR status, activities in stores, sales status, pricing, consumer expectations, and requests to the government (see questionnaire sheet in Appendix B).

## 5. Results

### 5.1. Producer Attitudes and Trends

5.1.1. Motivations and Difficulties in Obtaining Certification

All seven target fishers gave valid responses to the questionnaire (Figure 1).

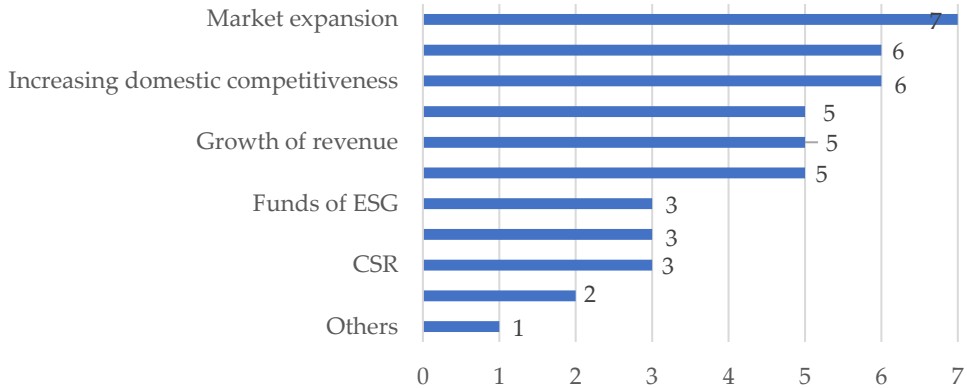

**Figure 1.** Reasons for acquiring MSC certification (multiple-choice answers, *n* = 7).

The majority indicated that their purpose for obtaining MSC certification was to increase both domestic and international products, including market expansion (all companies), increasing domestic competitiveness (six companies), and promoting exports (five companies). The respondents also noted a strong desire to contribute to sustainability

(six companies). We noted the high aspirations of certified fishers to focus on sustainability while simultaneously growing their own companies. Suzuki stated that the goal of the international certification of fishery products in Japan is to shift domestic fisheries toward sustainability, in line with international standards, and to increase global competitiveness [44]. This survey results showed that the producers recognized both values.

The difficulties they encountered in obtaining certification included the complexity of the procedures (six companies), financing (five companies), and language barriers (four companies) (Figure 2). In addition to the cost and time burdens, the complex English language documentation and communication with the head office were considerable burdens for applicants.

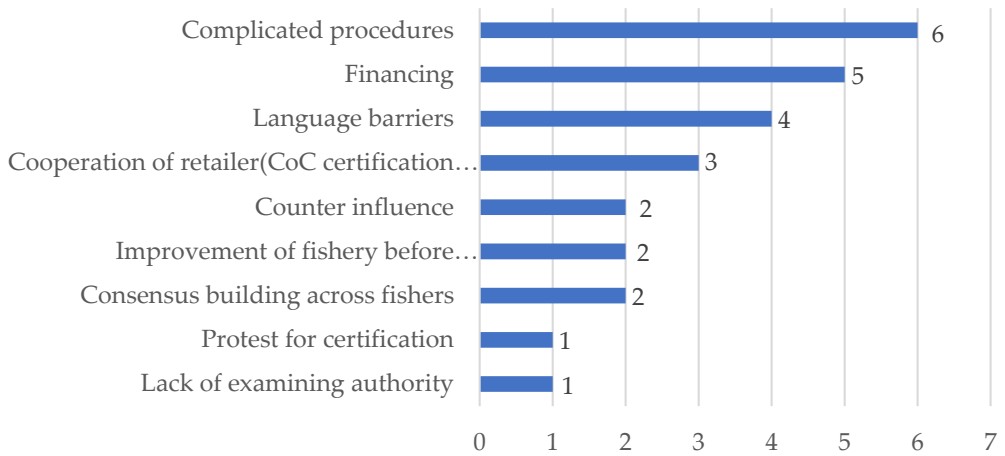

**Figure 2.** Difficulties experienced in obtaining MSC certification (multiple-choice answers, *n* = 7).

5.1.2. Effects of Holding Certification and Influences of International Official Events

As shown in Figure 3, few fisheries (two companies) received a positive response from consumers regarding certification. Conversely, many fisheries (five companies) received a direct response from clients, and their business partners. Only one company stated that Tokyo 2020 was one of the triggers to obtaining certification in terms of the impact of social conditions, while the majority (four companies) acknowledged the impact of the SDGs (Figure 4). Multiple fisheries cited the low price of MSC products as well as the lack of domestic sales channels and the need to raise awareness of MSC certification in the open-ended responses regarding the challenges with MSC certification.

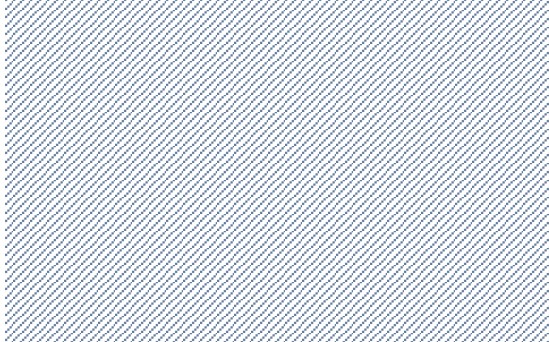

**Figure 3.** Response to MSC certification from clients and consumers (single answer, *n* = 7).

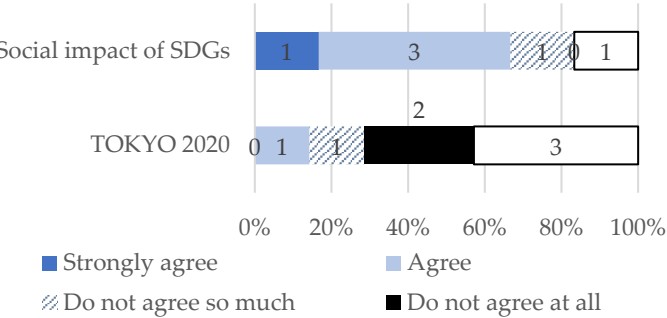

**Figure 4.** Triggers for obtaining MSC certification (single answer, *n* = 7).

The results of the surveys administered by the Ministry of Agriculture, Forestry and Fisheries (MAFF) in 2017 and 2020 [45,46] showed no change in those three years for the following (Figure 5a): reasons for wanting to obtain fishery certification in Japan: "to improve the image to differentiate and add value" (approx. 70%) and "to promote sustainability to consumers" (approx. 50%). Only approx. 20% of respondents wanted to "increase revenues by expanding exports" (with multiple responses). The results of this survey demonstrated the difference between MSC-certified fisheries and fisheries in general. Whereas MSC-certified fisheries aimed at more practical benefits, general fisheries had conceptual objectives, such as image and added value.

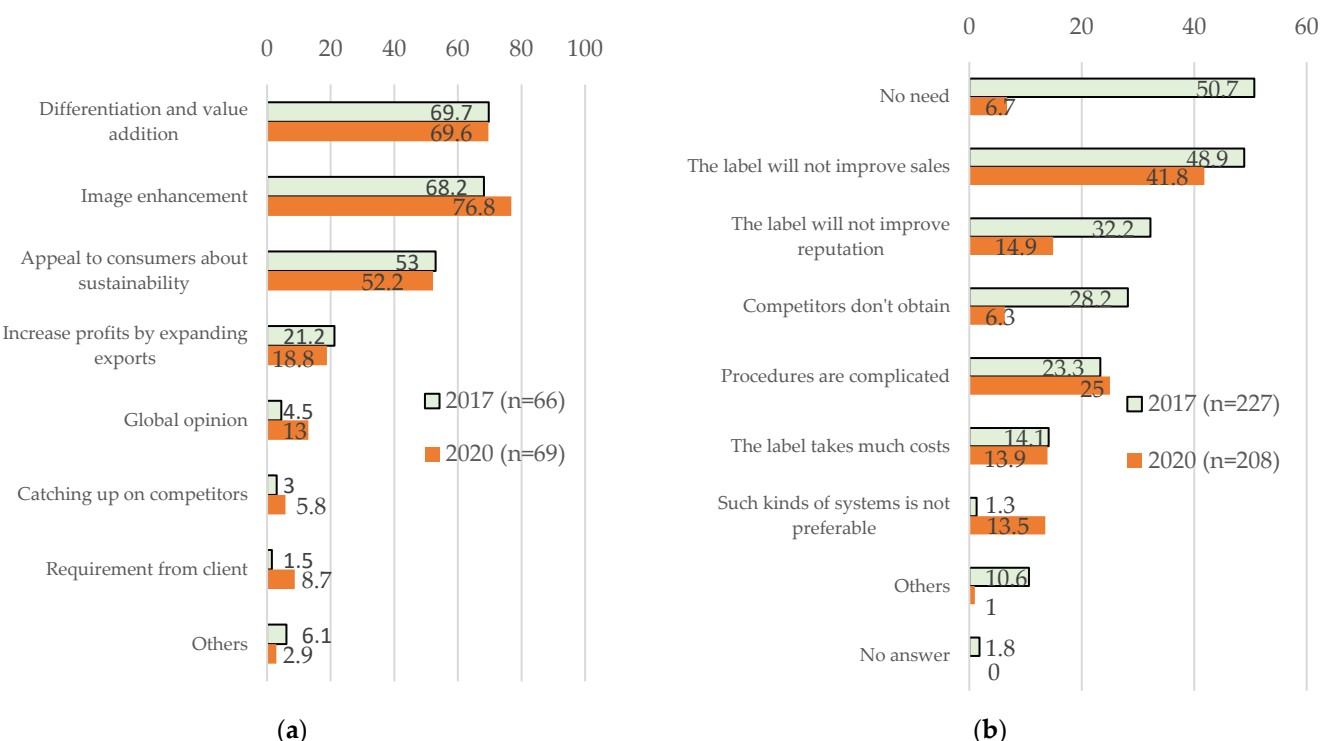

**Figure 5.** Results of surveys administered by MAFF regarding fishery certification (based on MAFF surveys, drawn by the authors): (**a**) reasons for wanting to obtain fisheries' certification; (**b**) reasons for not wanting to obtain fisheries' certification (%, multiple answers).

A substantial change was identified over those three years in the reasons for not wanting to obtain certification (Figure 5b): in 2017, "no need" was the top answer, accounting for approx. 50%, but in 2020, this answer dropped to sixth place, with a proportion of only 6.7%. This result indicated that awareness of the need for fisheries' certification rapidly increased over the three-year period. However, more than 40% of the respondents in both groups (2017 and 2020) did not think that acquiring certification would improve sales, with

this answer ranking second in 2017 and first in 2020. In summary, although awareness of the need for certification is improving, the price premium is not being realized.

### 5.2. Distributor Attitudes and Trends

5.2.1. Respondents and Their Industries

Of the 322 CoC holders since 2016, 98 provided valid responses (30.7% response rate). Of these, 76% were dedicated to seafood supply, 41% were seafood manufacturers, 35% were seafood wholesalers, and the rest were restaurants (13%) and general trading companies (4%) (Figure 6).

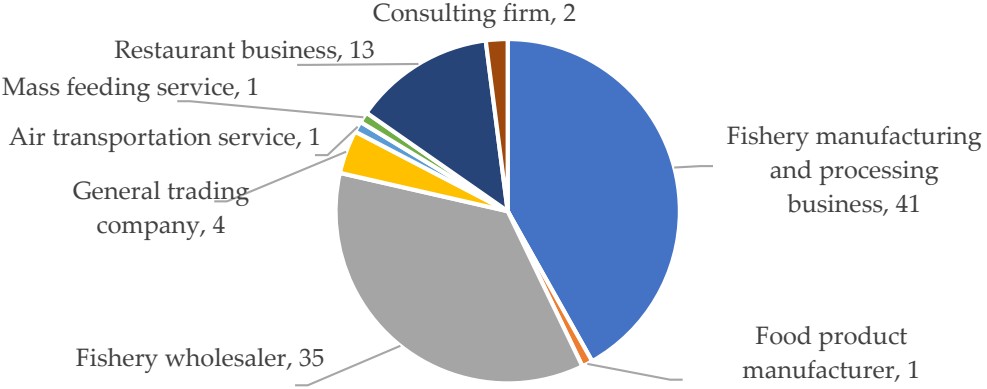

**Figure 6.** Industries of respondents with CoC (single answer, *n* = 98).

5.2.2. Motivations for, and Difficulties in, Obtaining Certification, and Impact of Social Conditions

Approx. 70% responded that they obtained CoC because of a request from their business partners, and approx. 50% wanted to contribute to seafood sustainability. Approx. 44% expected to expand their market, approx. 38% expected to improve their company image, and 33% wanted to make a favorable impression on consumers.

The most common difficulties encountered in obtaining certification were improving the internal system before the audit (approx. 40%) and the complexity of the procedures (approx. 40%), followed by internal consensus building (25%), with a tendency to opt for work within the organization rather than external matters.

In terms of the impact of social conditions, approx. 72% disagreed that they had much impact: approx. 49% strongly disagreed with Tokyo 2020 being a trigger for certification and approx. 12% agreed that Tokyo 2020 did not have much of an impact on promoting sustainable seafood certification. Conversely, almost the same number of organizations said that the social impact of the SDGs either had or had not triggered their activities (approx. 30% each; Figure 7).

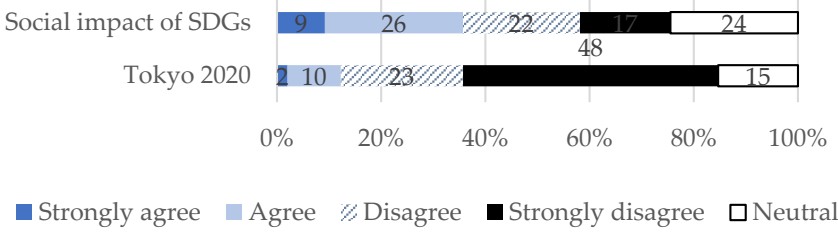

**Figure 7.** Triggers for obtaining CoC (single answer, *n* = 98).

### 5.2.3. Actual Sales of Sustainable Seafood

Regarding the sales of certified products (Figure 8a), approx. 50% of the companies were selling from one to five certified products. However, more than 40% of the operators did not always sell these products, indicating that certified product distribution was inactive. The purchase prices of the certified products were higher than those of similar products without certification in approx. 50% of the businesses (Figure 8b). However, approx. 70% of the distributors did not raise the sales price, suggesting that the cost burden for distribution and sales was added to the certification cost.

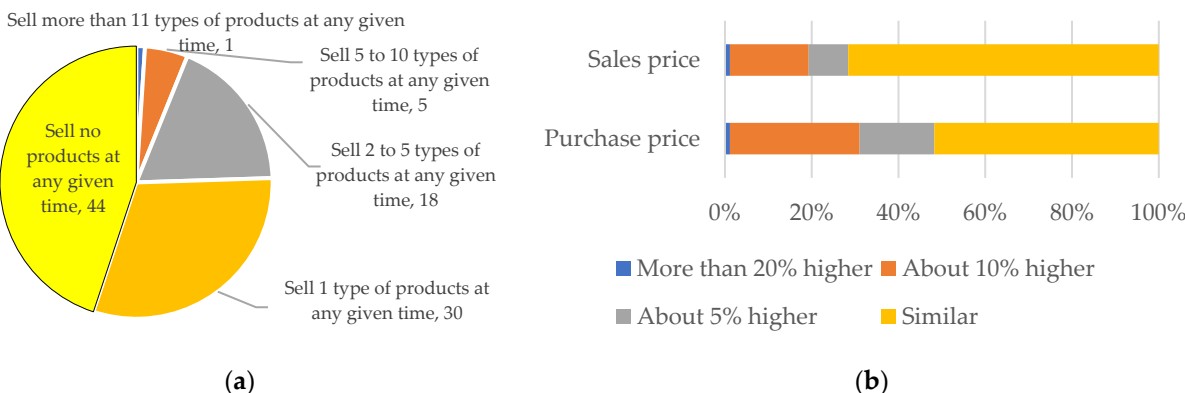

(**a**)            (**b**)

**Figure 8.** Actual sales of sustainable seafood: (**a**) number of certified products; (**b**) sales and purchase price of certified products (%, single answer, *n* = 98).

### 5.2.4. Effects of Certification, Communication, and Future Challenges

CoC certification holders found that consumers' response to certification was weak. Approx. 60% were uncertain (no response or other) about the consumers' response, approx. 20% experienced a reaction, and 20% experienced no reaction. Approx. 40% observed a difference in the reaction compared to their business partners (Figure 9a).

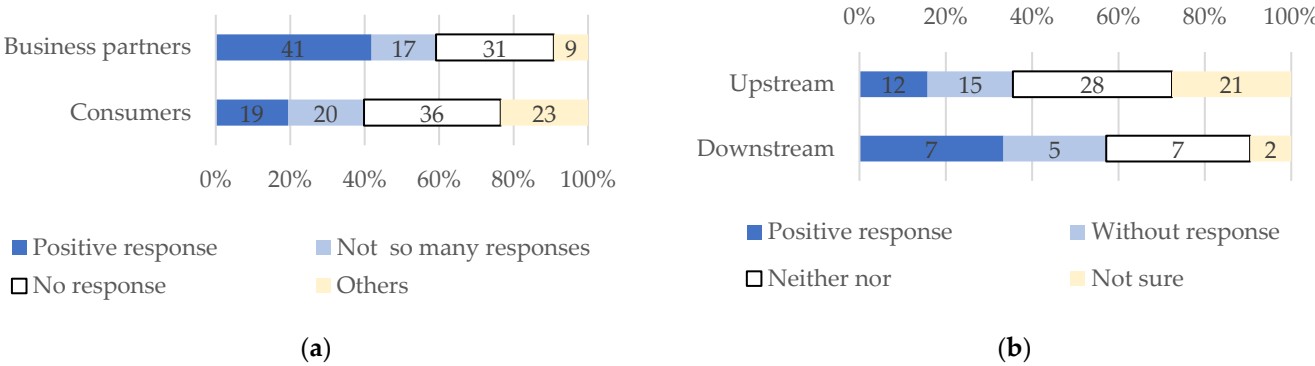

(**a**)            (**b**)

**Figure 9.** The response from business partners and consumers: (**a**) total average (single answer, *n* = 98); (**b**) difference in the distance from the consumers (single answer).

We categorized distributors into two groups in this survey, depending on their distance from consumers: the manufacturing and processing industry and food manufacturers and wholesalers were classified as upstream, and the rest as downstream. As shown in Figure 9b, downstream operators, who were closer to consumers, were more responsive, indicating that they experienced a certain level of impact. A Welch t test showed a significant difference between two groups (t(32) = 2.27, *p* = 0.029). However, nearly 70% of them observed no response, suggesting that the suppliers did not sufficiently communicate with consumers and that consumer knowledge of certification has not progressed.

Regarding consumer expectations, more than 70% of the distributors chose to raise awareness of sustainability and MSC or ASC certification. Preferential purchasing was the

answer for just under 50% of respondents, price premium the answer of approximately 30%, and looking for and buying products with the certification mark for only 10%, suggesting that the distributors understand that the awareness of the Japanese market needs to be raised before proactive purchase behavior occurs.

Nearly 50% of the distributors did nothing to communicate with consumers (Table 2). Even among those who did, 60% used only one method, such as displaying information on their websites, revealing that, while they hope to raise awareness and recognition, they are not actively engaging in outreach work. In addition, 70% of distributors did nothing to promote their products other than through their websites (Table 3). However, some operators actively took initiatives, such as developing original menus and providing education.

**Table 2.** Communication with consumers on sustainable seafood (*n* = 98).

| Communication | Number of Respondents | Percentage for All Respondents | Percentage for Respondents with Communication |
|---|---|---|---|
| Communication | 46 | 47% | |
| No communication | 52 | 53% | |
| Only one method | 31 | 32% | 60% |
| Only Internet | 31 | 32% | 60% |
| Only on-site | 13 | 13% | 25% |
| Combination of Internet and on-site | 8 | 8% | 15% |

**Table 3.** Promotion of their sustainable seafood products.

| Promotions Made | Number of Respondents | % |
|---|---|---|
| Nothing in particular | 70 | 71% |
| Sale of original products and menus | 15 | 15% |
| Educational activities inside and outside store | 13 | 13% |
| Sales fair with expanded sales floor | 8 | 8% |

Analyzing the relationship between transmission to consumers and response (Figure 10), we found that operators who transmitted information to consumers were three times more likely to experience a response than those who did not. In particular, the response received by businesses that transmitted messages through the Internet as well as in the field was more than 40%, compared with approximately 10% of those that did not transmit information. This finding indicated that efforts to transmit information led to a real sense of effectiveness.

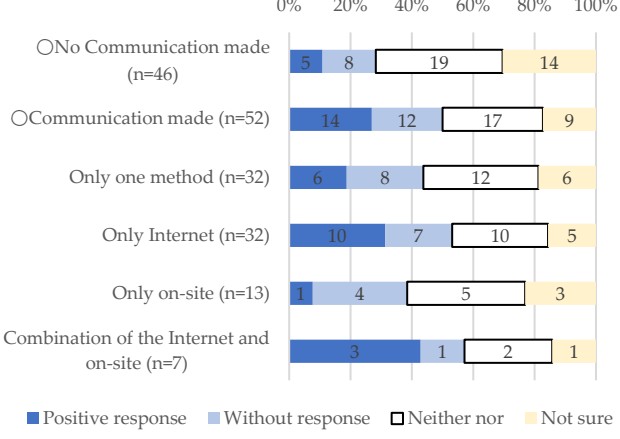

**Figure 10.** Relationship between information transmission to consumers and response.

Considering the relationship between other approaches to consumers and responses (Figure 11), we found that the response of distributors that directly communicated with consumers, such as by providing education inside and outside the shop, was greater, at over 40%. Conversely, of the eight businesses that held sales fairs with an extended sales area, one company found responses, while two did not.

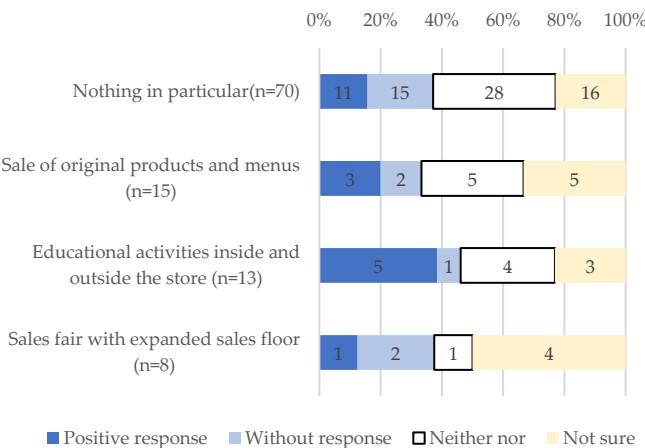

**Figure 11.** Relationship between other approaches to consumers and response.

When asked whether consumers generally feel that their awareness of sustainability and environmental considerations is increasing, overall opinions were divided into just two groups (Figure 12). Approximately 70% of downstream distributors felt that their awareness was improving, which was remarkably different from the response of those upstream. A Welch t test showed a significant difference between two groups (t(35) = 2.19, $p = 0.034$).

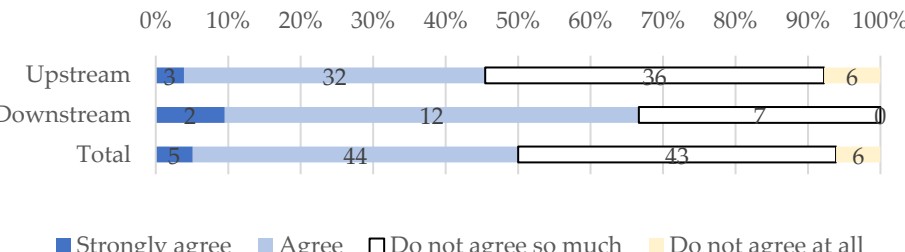

**Figure 12.** Consumer awareness of sustainability and environmental considerations is increasing (single answer).

The survey asked about requests to the government (Figure 13). To lead a campaign was the most common request, followed by subsidies, with both being made by approximately 50% of distributors. This shows that the distributors are expecting the government to lead and are willing to collaborate with political sectors. This survey also showed that cost-based challenges exist when distributing and selling certified products. Although fisheries can receive substantial subsidies [47], public funds for distributors are limited. Although some distributors were working on communicating with consumers, the response had not been sufficient, and so a demand for the government to build momentum was noted.

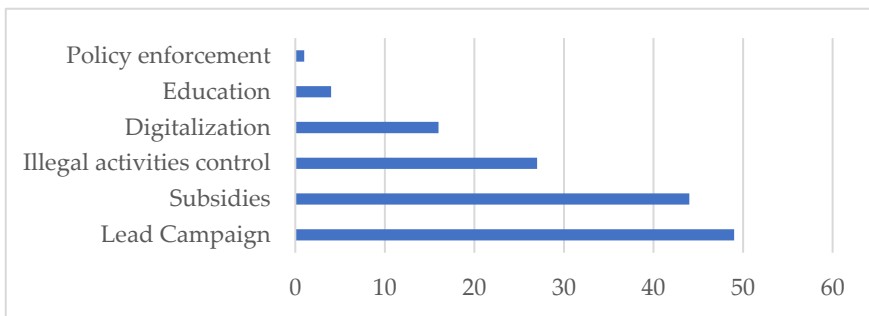

**Figure 13.** Request for the government from the distributors (multiple answers, *n* = 98).

The questionnaire lastly asked respondents to freely answer regarding what is needed to encourage consumers to choose sustainable seafood (Figure 14). A total of 68 out of 98 answered (69%). Education was the top demand, followed by increased awareness and media strategies; all these responses are related to raising awareness. Secondly, price adjustments and raising consumer income were mentioned, which shows that price is the second most important issue. Resource management, collaboration with the government, and the provision of subsidies and incentives are requests for government collaboration. Both Figures 13 and 14 show that collaboration with the government is wanted by the supply chain stakeholders.

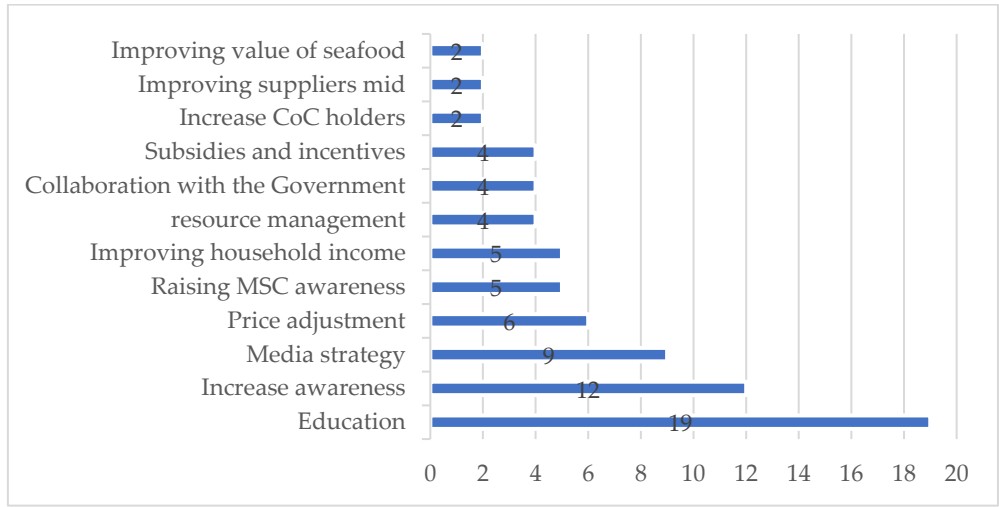

**Figure 14.** Need to encourage consumers to make sustainable seafood choices (free style, multiple answers, *n* = 98).

## 6. Discussion

### 6.1. Further Adopting Sustainability in the Supply Chain in Japan

The survey findings support our hypothesis, as differences were found in the level of awareness of proactive producers and distributors. A lack of consistent, mutual distribution of sustainability information was also found among these stakeholders. We then found the factors that influence these differences and inconsistencies.

Firstly, MSC-certified fisheries are proactively working toward sustainability. They are aware of the need to increase sustainability and expand sales channels. They are highly motivated, and have overcome various obstacles, such as high costs, complicated procedures, and language barriers, to obtain fishery certification. These obstacles are the hurdles facing the fisheries: only 12 fisheries have been certified as of 2022, and increasing the number of certified fishers is a challenge. Approximately 3000 small-scale regional fisheries account for a large proportion of Japanese fisheries [48], and they may regard the obstacles as too great. However, MSC has recently encouraged multiple fisheries in

the same waters to jointly apply, sharing the costs and obtaining certification as a group. The Kochi and Miyazaki Preparatory Council for International Certification of Pelagic Single Fishing Fishery for Albacore and Skipjack Fishing, which is an aggregate of two fishing cooperatives in two prefectures with 18 fishing vessels for albacore and skipjack, is a successful example of this in Japan [49]. As such, the ingenuity and efforts of the fisheries and certificate scheme owners should help to ensure that certification is attainable in a more realistic way, and is inclusive of small-scale fisheries.

Secondly, the survey found that the majority of CoC-certified distributors were passive in obtaining certification, doing so at the request of their business clients, and were generally inactive, being little involved in the dissemination of information or raising consumer awareness. This finding indicated that the CoC certification is not being fully used, despite the additional cost, and that cost is not being recovered. The respondents noted no confrontation or pressure from opposing forces, such as particular environmental NGOs or fishery organizations, in obtaining the certification, unlike the case of fishery certification. Most organizations experienced no difficulties in obtaining certification, and fewer hurdles than were experienced for fisheries' certification. Challenging CoC certification costs several hundred thousand yen, and only about one month is needed for recognition, whereas fisheries pay approximately 10–30 million Japanese Yen for assessment and face a 2–3-year research period [50]. However, this survey found hurdles within the organization: internal consensus-building remained difficult.

The downstream stakeholders have direct access to consumers, and feel more strongly about raising consumer awareness, so they should interactively disseminate information between consumers and upstream stakeholders. Businesses that proactively reach out to consumers are more likely to be responsive to consumer needs, indicating the importance of sharing best practices.

Thirdly, both producers and distributors were aware of the growing knowledge of SDGs among society and have high expectations for future consumer awareness of sustainable seafood, as well as for government campaigns and expanded education in general supporting the movement. As shown in Figures 1 and 7, stakeholders did not have much interest in participating in Tokyo 2020. Tokyo 2020 provided an excellent opportunity to follow the success of the London Olympics in 2012 in terms of introducing sustainable seafood sourcing codes. However, several studies criticized the Tokyo 2020 sustainable sourcing code, which allowed for unsustainable seafood to be procured [51–54]. The quality of the sustainable sourcing code for future international events held in Japan, such as the Osaka-Kansai Expo 2025, should be improved to ensure that international standards are achieved to help raise the supply chain and consumer awareness.

Finally, one of the challenges is continuing the certifications. MSC and CoC certifications require renewal after five years for fisheries and three years for CoC. In certain cases, the continuation of certification may be abandoned if the motivation is passive or decreases, a revenue benefit is not realized, or the financial burden is too large. At that point, an incentive for continued application would be increased demand. Stimulating consumer demand and verifying its effectiveness are urgent issues.

*6.2. Next Steps for Promoting Sustainable Seafood in Japan*

Additionally, for how long should we rely on fishery certification?

In the future, fishery certificates should not be the only solution to achieve a sustainable seafood supply. In the EU, a catch certification scheme was enacted in 2008 for all seafood products, enabling the traceability of all catches. In the USA, catch certification became compulsory for 13 major species in 2013, and a bill to make it applicable to all fish species has passed the House of Representatives. In Japan, the new anti-IUU law will be implemented in December 2022 to mandate catch documentation. The first phase of the law applies to abalone, sea cucumber, and glass eel for Type 1 (domestic fisheries) and squid, saury, mackerel, and sardine for Type 2 (imported products); a review is said to be scheduled every two years to eventually cover all seafood products. It is vital that supply chain

stakeholders will proactively implement this new rule and ask for a sooner application for more species, preferably all species. The operation of a catch certification system would eliminate IUU fishery-origin seafood from the market, provide traceability, and help to ensure a certain level of sustainability. The survey found a need to encourage supply chain stakeholders to be proactive and adopt this new regulation.

A future challenge is to improve the official definition of sustainable seafood. Tokyo 2020 resulted in at least 95% of the seafood being supplied to the main dining hall being imported fishery-certificate-certified products, owing to the supplier's decision, but this means that the percentage of domestic products was less than 5% [55]. There was a dilemma: by placing the highest priority on certified products, domestic products were not adopted, and although the sustainability of marine products was ensured, the associated problem of carbon dioxide emissions due to long-distance transportation remained. As shown in Chapter 2, current fishery certification does not include food mileage, carbon footprint, or gender equality in its sustainability requirements. As time progresses, the composite elements of the SDGs conditions will need to be achieved. To resolve these issues, as this survey suggests, the stakeholders must collaborate with the government to support supply chain stakeholders' efforts and implement transformative ocean science.

## 7. Conclusions

The challenge in Japan is increasing the sustainability of most seafood-supply stakeholders, particularly small-scale fishermen, who make up the majority of the fishing industry. The surveys revealed that this issue could benefit from increased awareness and the sustainable distribution of the supply chain that distributes the catch. What if the distributors become more well-educated, well-informed and well-motivated, proactively select sustainable seafood, and start to identify the sustainability of small-scale fishers to support local businesses? What if those suppliers could function as judges of sustainability and sell sustainable products, providing fuller information to the consumers? This survey sampled proactive, eligible stakeholders, and the next step is to focus on the majority of stakeholders, who are not certification holders and consumers.

The limitation of this survey is that the stakeholders who are non-proactive in sustainable seafood consumption do not even answer the survey. For this study, 100% of MSC certified fishers responded to the survey, but the answers from the CoC-certified stakeholders were 30.4%. The even lower rate of the answers to the questionnaires from the majority of non-certified stakeholders is predictable, and it may not be easy to survey the status of sustainability-averse stakeholders. The possible extension is once the stakeholders realize that protecting the ocean protects their income and future. It is also important to obtain primary data on consumer awareness and behavior. Achieving a sustainable blue economy in Japan and many other Asia-Pacific countries is crucial to achieving sustainability in the seafood supply chain globally.

In addition to SDG14 ("Life below water") for sustainable fisheries, synergy effects of multiple SDGs for sustainable production and consumption patterns must be ensured, as pursued in SDG12 ("Responsible consumption and production"), SDG13 ("Climate action") and SDG5 ("Gender equality"). The United Nations has declared this the Decade of Ocean Science, setting this goal for 2030. This era provides a profound opportunity to achieve a nexus of science with policy, business, and consumption. The successful implementation of transformative ocean science in society is, therefore, expected.

**Author Contributions:** Conceptualization, M.I.; methodology, formal analysis, data curation, M.I., M.A.; writing—original draft preparation, M.I.; writing—review and editing, M.M., M.A.; visualization, M.A., M.I.; supervision, M.M.; project administration, M.I. All authors have read and agreed to the published version of the manuscript.

**Funding:** This study did not directly receive funding, but the Blue Seafood Guide project, including its development and dissemination, was supported by The David and Lucile Packard Foundation, USA (grant numbers 2013-38894, 2014-39955, 2014-40240, 2015-63093, 2015-62480, 2016-64312, 2017-66558, 2018-67814, 2018-68096, 2019-68972, 2019-68987, and 2020-71494) and The Walton Family Foundation, USA (grant numbers 2016-1906, 2017-1066, 2018-168, and 00101942).

**Acknowledgments:** We express our sincere appreciation to Atsushi Sunami, National Graduate Institute for Policy Studies; Yuta Ando, Kyoto University; Kozo Ishii, MSC Japan; Koji Yamamoto, ASC; and all the certified stakeholders who kindly answered the questionnaires, for their valuable support.

**Conflicts of Interest:** The authors declare no conflict of interest.

## Appendix A

**Questionnaire for MSC-certified fishers** (original version was in Japanese).
Methodology:       Google Form was used
Response period:   14–24 January 2022
**Requested Text:**
We would like to ask your favor to answer the questionnaire from those who have obtained MSC certification for the research at Kyoto University Graduate School of Global Environmental Studies. The result will be used for research and thesis purposes at the Kyoto University Graduate School of Global Environmental Studies. The results of your responses will be anonymous and your company name will not be linked to your answers. Thank you in advance for your cooperation.
Kyoto University Graduate School of Global Environmental Studies
Minako Iue, PhD candidate
minakoiue@icloud.com 09083824794
**Entry Requirements**
Name of Fishery Certification Organization
Name and title of person filling out the questionnaire
Contact telephone number
E-mail address
**Questions**

1. What is the purpose of your challenge to obtain MSC certification? Please check all that apply from the following. Please indicate any others. (Multiple answers allowed)

   To promote sustainability to consumers
   Want to contribute to sustainability
   Expect to increase revenue
   Expect to promote exports
   Expect to improve CSR
   Expect to improve ESG
   Expect to improve corporate image
   Expect to participate in the Tokyo 2020 Olympic and Paralympic Games
   Enhancement of domestic competitiveness
   Market expansion
   Other

2. Why did you choose MSC certification over other certifications? Please check all that apply. Please indicate any other reasons. (Multiple answers allowed)

   Internationality
   Credibility
   Recognition
   Cost-effectiveness
   Price
   Other

3. What difficulties did you encounter in obtaining certification? Please check all that apply. (Multiple answers are possible.) Please list any others.

Financing
Complicated procedures
Consensus building among fishermen
Cooperation of distributors (e.g., CoC certification)
Improvement of fishery prior to assessment
Confrontation with opposing forces
Pressure or opposition to certification
Language barriers (e.g., English)
Others

4. Was there a difference in the reaction from consumers before and after certification? Please select the appropriate answer.

Yes · No · Neither · Do not know

5. Did you notice any difference in reactions from suppliers, retailers, restaurants, etc. before and after certification? Please select the appropriate answer.

Yes · No · Neither · Do not know

6. Did the Tokyo 2020 Olympic and Paralympic Games trigger the acquisition of certification? Please select the appropriate answer.

Not at all
No, not at all
Cannot say either way
Strongly agree
Strongly agree

7. Did the social impact of the SDGs trigger your decision to obtain certification? Please select the appropriate answer.

Not at all
No, I do not think so.
Cannot say either way
Strongly agree
Strongly agree

8. What are your positive and negative aspects of the certification?

(Descriptive answers)

9. What do you think are the challenges for MSC certification in the future?

(Descriptive answer)

**Appendix B**

**Questionnaire for CoC Certificate holders** (original version was in Japanese)
Methodology:        Google Form was used
Response period:    February 17–March 19, 2022
**Requested Text:**
We would like to ask your favor to answer the questionnaire from those who have obtained MSC/ASC Chain of Custody certification for the research at Kyoto University Graduate School of Global Environmental Studies. This is an academic contribution to the development of sustainable fisheries by considering the opinions of those involved in the fisheries industry who are leading the way in the production, distribution, and sustainable seafood consumption. The research will be used for the dissertation at Kyoto University Graduate School. The results of your responses will be anonymous, and your company

name will not be linked to your answers and will not be made public. Thank you very much in advance for your cooperation.

Kyoto University Graduate School of Global Environmental Studies
Minako Iue PhD candidate
minakoiue@icloud.com 09083824794
**Entry Requirements**
Name of CoC certified organization
Type of Business
Type
Name and title of person filling out questionnaire
Contact telephone number
E-mail address
**Questions**

1.  What is the purpose of your challenge to obtain CoC certification? Please check all that apply. Please indicate any others. (Multiple answers allowed)

    To promote sustainability to consumers
    Want to contribute to sustainability
    Expect to increase revenue
    Expect to promote exports
    Expect to improve CSR
    Expect to improve ESG
    Expect to improve corporate image
    Wanted to participate in the Tokyo 2020 Olympic and Paralympic Games
    Increase domestic competitiveness
    Market expansion
    To respond to requests from business partners
    To obtain subsidies
    Other

2.  Why did you choose MSC/ASC Chain of Custody certification over other certifications? Please check all that apply. Please indicate any other reasons. (Multiple answers are acceptable.)

    Internationality
    Credibility
    Recognition
    Cost-effectiveness
    Price
    Requests from suppliers
    Prospects
    Recommendation from NGOs, etc.
    Other

3.  What difficulties did you encounter in obtaining certification? Please check all that apply. Please indicate any others. (Multiple answers allowed)

    Financing
    Complicated procedures
    Internal consensus building
    Cooperation of distributors (e.g., CoC certification)
    Improvement of internal system prior to audit
    Confronting Opposing Forces
    Pressure or opposition to certification
    Language barriers (e.g., English)
    Others

4. Was there a difference in the reaction from consumers before and after certification? Please check all that apply.

   Yes · No · Neither · Do not know

5. Was there any difference in the reaction from suppliers, retailers, restaurants, etc.

   before and after obtaining the certification? Please check all that apply.
   Yes · No · Neither · Do not know

6. Did the Tokyo 2020 Olympic and Paralympic Games trigger your certification? Please check all that apply.

   Strongly disagree · Disagree · Neither · Agree · Strongly agree

7. Did the social impact of the SDGs trigger your decision to obtain certification? Please check all that apply.

   Strongly disagree · Disagree · Neither · Agree · Strongly agree

8. What are your positive and negative aspects of the certification?

   (Descriptive)

9. What do you think are the challenges for CoC certification in the future?

   (Descriptive)

10. What media do you use to communicate to consumers about your CoC-certification? Please check all that apply. Please indicate any others. (Multiple answers allowed)

    On our website
    SNS (Facebook, Instagram, Twitter, etc.)
    In-store POP, etc.
    Newspaper ads, etc.
    Flyers, etc.
    Not specifically communicated

11. What else do you do to reach out to consumers regarding CoC-certified products? Please check all that apply. Please indicate any others (multiple answers allowed).

    Sales fairs with an expanded sales floor
    Sale of products at reduced prices
    Sales of original products or menus
    Educational activities inside and outside the store
    Nothing in particular
    Other

12. We would like to ask you about the status of sales of certified products. Do you always sell certified products? If so, how many species do you sell?

    No, I do not sell them all the time.
    Always sell about 1 type of fish
    2 to 5 kinds of products are always sold.
    5 to 10 species are sold at any given time
    11 or more types of fish are sold at any given time

13. How do the purchase prices of certified products compare to similar products that are not certified?

    About 10% higher
    About 5% higher
    About the same
    5% lower
    10% cheaper
    Other

14. How does the selling price of certified products compare to similar products that are not certified?

    About 10% higher
    About 5% higher
    About the same
    5% lower
    10% lower
    Other

15. What do you expect from consumers? Please check all that apply. Please indicate any others. (Multiple answers allowed)

    They will prioritize to purchase certified products
    They will pay more for certified products to cover the cost
    Raise awareness of sustainability of seafood
    Will look for and purchase MSC/ASC certified products online, etc.
    Would like to be aware of MSC/ASC certification
    Other

16. Have you received any reactions from consumers regarding certification?

    Yes very much · Yes · Not much · Not at all · Other

17. If you chose "Yes very much" or "Yes" in 16, what kind of reaction have you received from consumers? (Free description)

18. In addition to fishery certification, there are other indicators and systems that indicate the sustainability of fishery resources. Are you aware of any of the following programs? Please check all that you know. Please indicate any others you are aware of. (Multiple answers allowed)

    Japanese rating program "Blue Seafood Guide
    U.S. rating program "Seafood Watch
    Good Fish Guide" rating program in the U.K.
    European rating program "Mr. Good Fish
    I don't know anything about it.
    Other

19. Do you know that Blue Seafood Guide also introduces MSC/ASC certification?

    Know well, Know, Not much, Do not know at all, Other

20. Do you feel that consumers in general are becoming more aware of sustainability and environmental considerations?

    Feel strongly·Feel·Do not feel much·Do not feel at all

21. What do you expect the government to do to encourage consumers to choose sustainable seafood products? (Multiple answers allowed)

    Tightening of regulations
    Deregulation
    Crackdown on illegal activities
    Provide subsidies
    Digitalization
    Campaigns
    Other

22. What do you think is needed to encourage consumers to choose sustainable seafood? (Free answer)

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
