# Peer review of "Seafood Sustainability Supply Chain Trends and Challenges in Japan: Marine Stewardship Council Fisheries and Chain of Custody Certificates"

_sustainability, doi:10.3390/su142013523_

Round 1

Reviewer 1 Report (Previous Reviewer 2)

The article has been modified a lot according to the suggestion. Still, no quantitative analysis is shown, but this article's recent version may be published after some minor modifications. Highlight the research questions at the end of the introduction part. The questions should relate to the originality/novelty of the study. An author's contribution table must be added in the literature review section which should clearly show that the research gaps. Add the limitation of research and possible extension clearly at the end of the conclusion. Authors may use a separate paragraph for that.

Author Response

Reviewer 2 Report (Previous Reviewer 3)

The manuscript addresses the sustainability of seafood in Japan and identify the reason why the awareness of seafood sustainability is low in Japan and designed the actions necessary to increase the demand for sustainable. The manuscript has improved and can be accepted for publication if the resolution of all figures is improved.

Author Response

Reviewer 3 Report (New Reviewer)

The main issue of the article is close to reality. I believe that this kind research can be useful to deal with the problems of implementations of sustainable development. An article is logically organized and has clear conclusions. All facts are significant.

Author Response

Reviewer 4 Report (New Reviewer)

This study assess the Japanese fisheries’ product and, specifically, identifies the reason for the low awareness of seafood sustainability. The paper also argues the necessary actions to increase sustainable seafood consumption. 

The analysis is build on a survey to collect data from the proactive stakeholders in seafood supply to determine the current status of sustainable seafood sales. 

I like the topic and the paper which is very well written. I do not have many comments that can improve the current version of the paper. however, I only suggest to improve the section "conclusions"  highlighting the contribution of the article with respect to the existing literature and provide some potential future research developments.

Author Response

This manuscript is a resubmission of an earlier submission. The following is a list of the peer review reports and author responses from that submission.

Round 1

Reviewer 1 Report

The writing was good.  The merit was there.  But the writing in Consumer part was the weakest link.   Need to revise the Consumer part and focus more on other issues in the discussion.  See the attachment for more details.

Reviewer 2 Report

The article describes the challenges of the sustainable seafood supply chain in Japan. The orientation of the article is well but there is a lack of clarity in this research. First of all, no definition of sustainable seafood is provided clearly in the manuscript. Not all readers are aware of what sustainable seafood is. The entire article is based on some case studies and a review of the questionnaire. The methodology used in this research is poor and has not capable enough to obtain a valid decision. The approach used in obtaining strategic decisions is purely qualitative. I would recommend applying some quantitative approach so that the parameters could be identified properly. Also, a proper statistical approach should be utilized to analyze the collected data. Moreover, no concluding remarks are provided in the article. At the end of the article "Necessary steps for promoting sustainable seafood" is discussed but no vital remarks could be found. The entire discussion is also full of literature rather than a concrete result discussion or solution strategy.

Except for these major issues, the article also needs some minor amendments. The abstract section should be rewritten. It should be crisper. The elaboration is too long which can be reduced. Moreover, try not to use "I"/"We" in the abstract section. The literature lacks the context of the study and the research gap. Moderate English modification is required. The entire article needs adequate amendment. Therefore, I must reject this article for possible publication.

Reviewer 3 Report

The manuscript addresses the sustainability of seafood in Japan and identify the reason why the awareness of seafood sustainability is low in Japan and designed the actions necessary to increase the demand for sustainable. The manuscript can be accepted as long as the following points are addressed.

1-Abstract should include the major results.

2- The authors address; however, they did not propose a solution to the problem; they mention that “the sustainable seafood consumption may be achievable by enforcing interactive exchange of information and collaboration”. This solution is very preliminary and wide.

3- Acronyms and abbreviations should be declared at the first use, for instant, SDGs

4- Page 2, line 83: You need to insert a reference here “Hori et al. showed that an environmentally…”

5-n=7 is very small number to give such kind of concluding results.

6- Statistical analysis, correlations and tests should be conducted.

7-Section 3 is dedicated for results of this paper, however, the authors mentioned other works, it should be moved to the literature review section. In the discussion section, you can compare your results with others. The same in section 4-2

8- In section 4-2, the authors should give guidelines based on their findings not other findings. Again, you can compare your findings with others in the discussion section.

9- I recommend adding a conclusion section.

10-Structure and grammatical issues:

a.      “So” is informal in abstract

b.      Page 2, last paragraph: it is better to use present simple rather than past simple “Our purpose in this study was ……” – “Our hypothesis was ………..”